# Peer review of "Comparing diversity, negativity, and stereotypes in Chinese-language AI technologies: an investigation of Baidu, Ernie and Qwen"

_PeerJ Computer Science, doi:10.7717/peerj-cs.2694_

## Round 0.1 · original submission · Major Revisions

All 3 reviewers have requested significant revisions to your work. Please attend to all their comments in detail.

Reviewer 1 ·

Basic reporting

Summary:

In this paper, the authors examine social biases in Chinese Large Language Models (LLMs) and the Baidu search engine by analyzing their outputs for 240 social groups across 13 categories. The study focuses on biases in Ernie and Qwen, two leading Chinese LLMs, and compares them to Baidu. The findings reveal that Qwen shows more diversity in views but also generates more negative content compared to Ernie, which tends to produce safer outputs. Both LLMs and Baidu perpetuate stereotypes, with Qwen displaying a higher prevalence of offensive content. The research underscores the importance of promoting fairness and inclusivity in AI technologies, especially as they become more integrated into societal functions.

Strengths

1. The paper addresses a gap in fairness research by focusing on Chinese LLMs, which are often overlooked in existing work.

2. I expect that the paper would be of interest to the community and generate discussions.

Weakness:

1. While the authors mention various methods for measuring and mitigating bias in language models, the references provided are outdated, with the most recent articles being four years old (lines 42-43). Furthermore, not all the cited work focuses specifically on fairness in Large Language Models (LLMs), but rather on fairness in Language Models (LMs) in general. Although LLMs are built upon LMs, there are significant differences between the two, and more recent literature on LLM fairness should have been included.

2. Although the authors claim the unique cultural, social, and linguistic characteristics of the Chinese language, they do not provide a detailed discussion of the specific challenges that Chinese and other non-Western languages face with LLMs. It would have been helpful if the authors had provided concrete examples in the introduction to illustrate these challenges and their implications for fairness in LLMs.

3. The authors' approach appears to rely on the chain-of-thought methodology, but they fail to discuss how existing research uses chain-of-thought techniques to enhance and measure fairness in LLMs. Incorporating relevant work, such as:

Turpin, Miles, et al. "Language models don't always say what they think: unfaithful explanations in chain-of-thought prompting." Advances in Neural Information Processing Systems 36 (2024).

Wei, Jason, et al. "Chain-of-thought prompting elicits reasoning in large language models." Advances in neural information processing systems 35 (2022): 24824-24837.

Chu, Zhibo, Zichong Wang, and Wenbin Zhang. "Fairness in large language models: a taxonomic survey." ACM SIGKDD explorations newsletter 26.1 (2024): 34-48.

4. Some pictures are blurry, such as Figure 5 at the top.

Experimental design

Although the article describes the diversity analysis of bias detection and model output, the technical details of some important statistical analyses are under-described, such as how statistical tests of diversity and bias are handled.

Validity of the findings

While the paper discusses the issue of bias in Chinese AI models, it does not adequately emphasize its impact or innovation on the field. While the research is important, it may be difficult for readers to directly feel the unique contribution of this research in the field of AI bias, especially when compared to existing fair LLMs research.

Reviewer 2 ·

Basic reporting

This study examined Chinese language models including Baidu (auto-completion in this search engine), Ernie (Chinese-centric LLM), and Qwen (Chinese-centric LLM) in terms of diversity, negativity, and stereotypes on a dataset of 240 social groups across 13 categories describing Chinese society. In particular, this study prompted the LLMs for candidate words describing such groups.

Overall, this study proposed to examine a critical topic in the development of Chinese language models. The major suggestions are about the robustness of the experiments and the depth of the discussions.

Experimental design

(1) The study involved multiple comparisons across different models. To improve the robustness of the study, it is necessary to perform statistical tests with adjustment.
(2) The prompt templates are overall very similar. The findings may be biased toward these templates.
(3) It is recommended to include an additional experiments using all English with these three models to construct baseline results.

Validity of the findings

Please see my review on experimental design.

Additional comments

(4) Can the authors further discuss why Ernie generates less negative content? Was it because of the data it was trained on or the way it was trained?
(5) The findings regarding diversity may be trivial. Can the authors further discuss them?
(6) It would be better to add a vertical line at 0.5 in Figure 9.
(7) What do different color mean in Figure 11?
(8) Would the language tools used to evaluate these responses be biased?

Reviewer 3 ·

Basic reporting

The authors explore the diversity, negativity, and stereotypes in Chinese large language models (LLMs). Through extensive experiments, they arrived at interesting findings, i.e., LLMs can potentially provide more nuanced views, yet are not entirely free from reinforcing stereotypes.

Experimental design

The data collection and analysis processes are rigorous. The authors used a wide range of NLP techniques to arrive at the findings. My biggest concern is about the validity of the findings. Please see below.

Validity of the findings

I find several findings doubtful.

First, "Baidu and Qwen exhibit concerning levels of potentially offensive generated content
(1 out of 3 candidate words has a negative sentiment) compared to Ernie, which appears much safer (only approximately 1 out of 10 candidate words is negative)." Negative sentiments do not necessarily indicate "offense." I would expect at least some examples of offensive words or a qualitative analysis.

Second, "Overall, Ernie and Qwen share 26.52% and 27.81% of their completions with Baidu, meaning that 1 out of 3 candidate words generated by the LLMs to describe different social groups coincide with the views embedded in Chinese online search queries. We regard these overlapping completions as stereotypical." It's not accurate to regard overlaps with Baidu as stereotypical, which is a bit arbitrary. I'd suggest a qualitative inspection to support such statements.

---

## Round 0.2 · Minor Revisions

Dear authors,

Feedback from the reviewers is now available for your revised paper. It is still not recommended that your article be published in its current format. However, we strongly recommend that you address the minor issues raised by Reviewer 2 and resubmit your paper after making the necessary changes.

Best wishes,

Reviewer 2 ·

Basic reporting

The authors have addressed most of my concerns except for my question that "Can the authors further discuss why Ernie generates less negative content? Was it because of the data it was trained on or the way it was trained?"

The authors mentioned that Ernie and Qwen are both close-source. It is not true. They are both open-source models.

Experimental design

The authors have addressed most of my concerns.

Validity of the findings

The authors have addressed most of my concerns.

---

## Round 0.3 · accepted · Accept

Dear Authors,

Thank you for clearly addressing the reviewers' comments. Your manuscript now seems sufficiently improved and ready for publication.

Best wishes,

Reviewer 2 ·

Basic reporting

The authors have addressed my questions.

Experimental design

The authors have addressed my questions.

Validity of the findings

The authors have addressed my questions.

Additional comments

The authors have addressed my questions.